# OVOL2-Mediated ZEB1 Downregulation May Prevent Promotion of Actinic Keratosis to Cutaneous Squamous Cell Carcinoma

**DOI:** 10.3390/jcm9030618

**Published:** 2020-02-25

**Authors:** Maho Murata, Takamichi Ito, Yuka Tanaka, Kazuhiko Yamamura, Kazuhisa Furue, Masutaka Furue

**Affiliations:** 1Department of Dermatology, Graduate School of Medical Sciences, Kyushu University, Fukuoka 812-8582, Japan; muratama@dermatol.med.kyushu-u.ac.jp (M.M.); yukat53@med.kyushu-u.ac.jp (Y.T.); kazuhiko@dermatol.med.kyushu-u.ac.jp (K.Y.); kazfurue@dermatol.med.kyushu-u.ac.jp (K.F.); furue@dermatol.med.kyushu-u.ac.jp (M.F.); 2Research and Clinical Center for Yusho and Dioxin, Kyushu University Hospital, Fukuoka 812-8582, Japan; 3Division of Skin Surface Sensing, Department of Dermatology, Faculty of Medical Sciences, Kyushu University, Fukuoka 812-8582, Japan

**Keywords:** actinic keratosis, cutaneous squamous cell carcinoma, EMT, OVOL2, ZEB1, OVOL1, Twist1, Snail, vimentin, E-cadherin

## Abstract

Progression of actinic keratosis (AK) to cutaneous squamous cell carcinoma (cSCC) is rare. Most cases of AK remain as intraepidermal lesions, owing to the suppression of the epithelial-to-mesenchymal transition (EMT). Ovo-like transcriptional repressor 1 (OVOL1) and ovo-like zinc finger 2 (OVOL2) are important modulators of EMT in some tumors, but their roles in skin tumors remain elusive. This study elucidated the roles of OVOL1/2 in AK and cSCC using 30 AK/30 cSCC clinical samples, and an A431 human SCC cell line using immunohistochemistry and molecular biological approaches. Immunohistochemically, OVOL1/2 were upregulated in AK and downregulated in cSCC. Meanwhile, EMT-related factors, vimentin and zinc finger E-box binding homeobox 1 (ZEB1) were downregulated in AK and upregulated in cSCC. Moreover, ZEB1 expression was higher in tumors in which OVOL2 expression was low. Thus, we observed an inverse association between OVOL2 and ZEB1 expression in AK and cSCC. Although knockdown of OVOL1 or OVOL2 increased the mRNA and protein levels of ZEB1, only OVOL2 knockdown increased the invasive ability of A431. In conclusion, OVOL2 inhibits ZEB1 expression and may inhibit the promotion of AK into cSCC. OVOL2/ZEB1 axis may be a potential target for preventing the development of cSCC.

## 1. Introduction

Actinic keratosis (AK) and cutaneous squamous cell carcinoma (cSCC) are common types of precancerous and cancerous skin lesions, whose prevalence is increasing in the context of population aging [1]. A 2013 meta-analysis estimated that there were between 186,000 and 419,000 new cases of SCC (excluding SCC *in situ*) in the Caucasian population in the United States in 2012 [2]. AK and cSCC mainly arise on sun-exposed parts of the body in elderly people. Most cases of AK remain intraepidermal lesions and the progression of AK to cSCC is rare [3,4]. AK is a precursor of cSCC characterized by atypical keratinocytes within the epidermis. Early changes in AK start at the basal layer in the interfollicular epidermis [1,5]. Although the rate of transformation of AK to cSCC is low, 0.1% to 0.6% per lesion per year, consequences of advanced cSCC can be devastating [3,4]. Therefore, early-stage intervention and/or prophylaxis is essential. The transition from AK to cSCC is suggested to occur in one of two ways: through direct dermal invasion of the atypical cells found only in the basal layer of AK, or through secondary dermal invasion after the atypical cells have extended throughout the epidermis [5]. However, most AK cases are restricted to the epidermis for a long time, suggesting the existence of a potent mechanism that suppresses the epithelial-to-mesenchymal transition (EMT).

Mesenchymal and epithelial cell phenotypes are not completely irreversible [6]. For example, cells are thought to switch between these two states during embryonic development [6]. Moreover, EMT and the reverse process—mesenchymal-to-epithelial transition (MET)—are regarded as crucial processes in early organ development and wound healing [7,8]. In fact, a series of EMT and MET conversions could affect the generation of numerous adult tissues and organs. Besides normal tissue development, EMT plays important roles in stromal invasion by tumor cells [7]. During the EMT process, cells with an epithelial phenotype are considered to lose their apicobasal polarity, apical tight junctions, cell junctions, and cytoskeletal structures and subsequently acquire a mesenchymal phenotype [7,9,10]. The phenotype change between EMT and MET was recently reported to be important for several processes of cancer development [11,12]. Some signaling pathways and various molecules, such as zinc finger E-box binding homeobox 1 (ZEB1), Slug, Twist, and Snail, are involved in the progression of EMT. ZEB1 is a transcription factor located on human chromosome 10p11.22 and an essential regulator of EMT [13,14]. ZEB1 appears to play a critical role in the progression of malignant cancers including renal clear cell carcinoma, lung adenocarcinoma, breast cancer, and cervical squamous cell carcinoma [15]. In addition, ovo-like transcriptional repressor 1 (OVOL1) and ovo-like zinc finger 2 (OVOL2) have been identified as important molecules to regulate EMT [16,17].

In mammals, *OVOL1* and *OVOL2* are ubiquitously conserved genes that encode C2H2 zinc finger transcription factors [18,19]. OVOL1 and OVOL2 are important for the growth of epithelial tissue derived from reproductive cells in *Caenorhabditis elegans* and *Drosophila melanogaster* [18,20,21]. Of note, *OVOL2* might inhibit EMT in lung adenocarcinoma by transcriptionally suppressing an EMT-related gene, *Twist1* [22]. OVOL2 protein is considered an antagonist of TGF-β signaling that regulates EMT in breast cancer [23]. A relationship between OVOL2 and ZEB1 in osteosarcoma has also been reported [14]. However, the expression of OVOL1 and OVOL2 as well as their functions in skin neoplasms are poorly understood. Regarding cutaneous inflammatory diseases, Tsuji et al. demonstrated that the activation of aryl hydrocarbon receptor is associated with filaggrin and OVOL1 expression in atopic dermatitis [24]. Moreover, we recently showed that human epidermis and hair follicles express OVOL1 and OVOL2, and that OVOL1/2 are overexpressed in Bowen disease and downregulated in cSCC [21,25]. Therefore, in this study, we further examined the functional roles of OVOL1/2 in the transition from AK (cSCC precursor) to invasive cSCC. Using human clinical tumor samples and a human SCC cell line, we found that the OVOL2/ZEB1 axis was crucially involved in the development of cSCC. 

## 2. Materials and Methods

We conducted this study in accordance with the concepts enshrined in the Declaration of Helsinki. This study was approved by the Ethics Committee of Kyushu University Hospital (project number: 30-363, approved on 27th November 2018). All patients provided written informed consent to participate.

### 2.1. Tissue Samples

We examined 30 cSCC and 30 AK skin samples. All samples were taken from completely independent patients. Thirteen perilesional normal skin in AK patients served as control. All formalin-fixed (24 h in paraformaldehyde) and paraffin-embedded tissues were obtained from the archives of our hospital. At least three experienced dermatopathologists confirmed the diagnoses.

### 2.2. Immunohistochemical Analysis

Formalin-fixed, paraffin-embedded samples were cut into 4-μm-thick sections. We retrieved antigens using Heat Processor Solution pH 6 (Nichirei Biosciences, Tokyo, Japan) at 100°C for 40 min. We blocked nonspecific binding using a supernatant of 5% skimmed milk. The primary antibodies were diluted with Dako REAL Antibody Diluent (s2022; Dako Denmark A/S*,* Glostrup, Denmark). The primary antibodies used were rabbit anti-human OVOL1 (1:100, HPA003984; Atlas Antibodies, Bromma, Sweden), rabbit anti-human OVOL2 (1:100, NBP1-88754; Novus Biologicals, Littleton, CO, USA), rabbit anti-human ZEB1 (1:200, HPA027524; Atlas Antibodies), and mouse anti-human vimentin (prediluted by supplier, 722101; Nichirei Biosciences). We incubated sections with anti-OVOL1 overnight at 4 °C or with anti-OVOL2, anti-vimentin, or anti-ZEB1 for 2 h at room temperature. We then incubated sections with N-Histofine Simple Stain AP MULTI (414261; Nichirei Biosciences) secondary antibody for 30 min at room temperature. We detected immunoreactions using FastRed II (415261; Nichirei Biosciences) as a chromogenic substrate and counterstained with hematoxylin. Sections stained without primary antibody served as a negative control. Two independent dermatologists (M.M. and T.I.), who were blinded to the patients’ clinical information, performed immunohistochemical assessments. We captured images by microscopy (BX61VS; Olympus, Tokyo, Japan).

### 2.3. Cell Culture

We purchased a human SCC cell line, A431, from the American Type Culture Collection (CRL-1555; Manassas, VA, USA). We cultured cells in Dulbecco’s modified Eagle medium (D6429; Sigma-Aldrich, Tokyo, Japan) with a mixture of penicillin (100 units/mL), streptomycin (100 μg/mL), glutamine (29.2 mg/mL) (10378-016; Invitrogen, Carlsbad, CA, USA), and 5% fetal bovine serum (FBS) (CCP-FBS-BR; Cosmo Bio Co. Ltd., Tokyo, Japan) in 5% carbon dioxide (CO_2_) at 37°C. We passaged cells every 2–3 days until sub-confluence. 

### 2.4. A Gene-Specific Small Interfering RNA (siRNA) Transfection

The following pre-designed OVOL1- and OVOL2-specific siRNAs (Silencer Select; Life Technologies, Carlsbad, CA, USA) were used: OVOL1 (s9939, 5′-CAUAUCACCUCAUUUCUAAtt-3′) and OVOL2 (s33860, 5′-AGAUCGAAAAUCAAGUUCAtt-3′), as well as a control siRNA (4390846) as a non-targeting siRNA for transfection. We seeded the A431 cells into culture plates and transfected them with siRNAs using HiPerFect Transfection Reagent (301704; Qiagen, Hilden, Germany). 

### 2.5. Quantitative Reverse-Transcription Polymerase Chain Reaction (qRT-PCR) Analysis

A431 cells were seeded on 24-well plates (5 × 10^4^ cells per well) and were transfected with siRNAs (final concentration: 10 nM). We extracted total RNA using the RNeasy Mini Kit (74104; Qiagen) 48 h after siRNA transfection and used 2.5 μg RNA for reverse transcription with the PrimeScript RT reagent Kit (RR037A; Takara Bio Inc., Kusatsu, Japan). We performed qRT-PCR on a CFX Connect Real-time System (Bio-Rad Laboratories Inc., Hercules, CA, USA) using SYBR Premix Ex Taq (RR820S; Takara Bio Inc., Tokyo, Japan). We performed PCR amplification under the following conditions: 95 °C for 30 s, followed by 40 cycles of 95 °C for 5 s and 60 °C for 20 s. The levels of mRNA were normalized to that of β-actin. We used the ΔΔCt method to calculate the fold induction of each gene relative to that in the control group. We used the following primers (Hokkaido System Science Co., Sapporo, Japan): *OVOL1* forward, 5′-ACGATGCCCATCCACTACCTG-3′, *OVOL1* reverse, 5′-TTTCTGAGGTGCTGGTCATCATTC-3′; *OVOL2* forward, 5′-GGCAAGGGCTTCAACGACA-3′, *OVOL2* reverse, 5′-CTTCAGGTGGGACTCCAGAGA-3′; *E-cadherin* forward, 5′-TGCCCAGAAAATGAAAAAGG-3′, *E-cadherin* reverse, 5′-GTGTATGTGGCAATGCGTTC-3′; *vimentin* forward, 5′-GAGAACTTTGCCGTTGAAGC-3′, *vimentin* reverse, 5′-GCTTCCTGTAGGTGGCAATC-3′; *ZEB1* forward, 5′-GCACCTGAAGAGGACCAGAG-3′, *ZEB1* reverse, 5′-TGCATCTGGTGTTCCATTTT-3′; *ZEB2* forward, 5′-TTTCAGGGAGAATTGCTTGA-3′, *ZEB2* reverse, 5′-CACATGCATACATGCCACTC-3′; *Snail* forward, 5′-GCCTAGCGAGTGGTTCTTCT-3′, *Snail* reverse, 5′-TAGGGCTGCTGGAAGGTAAA-3′; *Twist1* forward, 5′-AAGGCATCACTATGGACTTTCTCT-3′, *Twist1* reverse, 5′-GCCAGTTTGATCCCAGTATTTT-3′; *β-actin* forward, 5′-ATTGCCGACAGGATGCAGA-3′, *β-actin* reverse, 5′-GAGTACTTGCGCTCAGGAGGA-3′. We conducted assays in triplicate and repeated them at least three times in separate experiments.

### 2.6. Western Blotting

A431 cells were seeded on six-well plates (2 × 10^5^ cells per well) and were transfected with siRNAs (final concentration: 10 nM). The cell proteins were extracted with lysis buffer (04719956001, Complete Lysis M; Roche Applied Science, Penzberg, Germany) after siRNA transfection. We collected lysates for OVOL1/2 protein detection 48 h post-transfection. Meanwhile, for vimentin, ZEB1, and E-cadherin protein detection, we collected lysates 72 h post-transfection. 

Protein levels were quantified with a BCA Protein Assay Kit (23227; Thermo Fisher Scientific, Waltham, MA, USA). We first dissolved the proteins in NuPAGE LDS Sample Buffer (NP0007; Invitrogen) and 10% NuPAGE Sample Reducing Agent (NP0009; Invitrogen) and heated them at 70°C for 10 min. We then separated the sample lysates using NuPAGE 4–12% Bis-Tris Protein Gels (NP0321BOX; Invitrogen) at 200 V, 200 W, and 180 mA for 23 min. The total amounts of protein loaded per well were 60 μg for OVOL1 and OVOL2, and 40 μg for vimentin, ZEB1, and E-cadherin. We subsequently transferred the sample proteins to polyvinylidene difluoride membranes (0.45-μm pore size, IPSN07852; Invitrogen). 

After blocking using WesternBreeze Blocker/Diluents A and B (46-7003 and 46-7004; Invitrogen), we probed the membranes with the following primary antibodies: mouse anti-human OVOL1 (1:100, ab74520; Abcam, Cambridge, UK), rabbit anti-human OVOL2 (1:500, ab83265; Abcam), rabbit anti-human ZEB1 (1:2000, NBP1-05987; Novus Biologicals), rabbit anti-human vimentin (1:1000, ab45939; Abcam), mouse anti-human E-cadherin (1:5000, 610181; BD Biosciences, San Jose, CA, USA), and rabbit anti-human β-actin as a housekeeping protein (1:1000, 3700; Cell Signaling Technology, Danvers, MA, USA). We diluted all primary antibodies with Signal Enhancer HIKARI for western blotting and ELISA (02272-741; Nacalai Tesque Inc., Kyoto, Japan) and incubated them with the membranes overnight at 4°C. We subsequently washed the membranes three times with Tris-buffered saline and Tween 20 (9997; Cell Signaling Technology) diluted into a 1:10 solution using distilled-deionized H_2_O (0.1% TBST). Next, we added anti-rabbit IgG (1:1500, 7074S; Cell Signaling Technology) or anti-mouse IgG (1:1500, 7076S; Cell Signaling Technology) secondary antibody diluted with 0.1% TBST to the membranes, followed by incubation for 30 min at room temperature. We then washed the membranes five times with 0.1% TBST. We detected protein expression using SuperSignal West Pico PLUS Chemiluminescent Substrate (34577; Thermo Fisher Scientific) or Chemi-Lumi One Super (02230-30; Nacalai Tesque Inc.) with the ChemiDoc touch imaging system (1708370; Bio-Rad Laboratories Inc.). We conducted all assays in triplicate and repeated them at least three times in separate experiments.

### 2.7. Matrigel Invasion Chamber Assay

Cell invasion assays were performed using Matrigel-coated Transwell cell culture inserts (8-μm pore size, 354480, Corning BioCoat Matrigel Invasion Chamber; Corning Inc., Corning, NY, USA) and control inserts without Matrigel coating (8-μm pore size, 353097, Falcon Cell Culture Insert; Corning Inc.). We seeded A431 cells onto six-well plates (2 × 10^5^ cells per well) and transfected them with control siRNA, OVOL1 siRNA, or OVOL2 siRNA for 48 h. We harvested the transfected cells and seeded them in the upper Transwell chambers (4 × 10^4^ cells) with serum-free culture medium. Meanwhile, we added culture medium supplemented with 5% FBS to the lower chambers. After incubation at 37°C in 5% CO_2_ for 22 h, we performed hematoxylin staining and counted the cells that translocated to the lower surface of the culture inserts. We counted the cells that penetrated the membrane at ×200 magnification (ECLIPSE 80i; Nikon, Tokyo, Japan) for all areas of the membrane. We conducted all assays in triplicate wells and repeated them at least three times in separate experiments.

### 2.8. Wound Healing Assay

A431 cells were transfected with control siRNA, OVOL1 siRNA, or OVOL2 siRNA and were seeded at 1 × 10^4^ cells per well on a 96-well ImageLock tissue culture microplate (4379; Essen Bioscience, Ann Arbor, MI, USA) pre-coated with type I collagen (637-00653, Cellmatrix Type I-A; Nitta Gelatin Inc., Osaka, Japan). Forty-eight hours post-transfection, we scratched the cell monolayers with a wound maker 96 (9600-0012; Essen Bioscience). The wound area of each well was automatically imaged every 2 h in a CO_2_ incubator using a live-cell imaging system (IncuCyteHD; Essen Bioscience). We measured the wound area relative to that at 0 h using IncuCyte software (9600-0012, a set of wound markers; Essen Bioscience). We conducted all assays by using 24 wells/condition and repeated them at least three times in separate experiments.

### 2.9. Proliferation Assay

A431 cells transfected with control siRNA, OVOL1 siRNA, or OVOL2 siRNA were seeded at 5000 cells per well on 96-well plates. After 48 h of incubation, we treated cells with the Premix WST-1 Cell Proliferation Assay System (MK400; Takara Bio Inc.) for 2 h. We performed spectrophotometry and measured the absorbance at 450 nm using a microplate reader (DTX 800; Beckman Coulter, Brea, CA, USA). We conducted all assays in triplicate wells and repeated them at least three times in separate experiments.

### 2.10. Colony Formation Assay

We performed colony formation assays using a Cytoselect 96-well cell transformation assay kit (CBA-135; Cell Biolabs Inc., San Diego, CA, USA) according to the manufacturer’s instructions. Briefly, we added a base agar matrix layer into each well of a sterile, 96-well, flat-bottomed microplate. We transfected the cells with siRNAs and seeded the transfected cells onto the base agar matrix layer (1 × 10^4^ cells per well). After 6–8 days of incubation at 37°C in 5% CO_2_, we confirmed colony formation under a microscope. To quantify the colony-forming cells, we solubilized colony-containing agar with Matrix Solubilization Solution and mixed it with MTT solution for 2–4 h at 37°C in 5% CO_2_. After confirming the formation of precipitate within the cells, we added detergent solution, followed by 2–4 h of incubation at room temperature in the dark. We measured the absorbance at 570 nm using a microplate reader. We conducted all assays in triplicate wells and repeated them at least three times in separate experiments.

### 2.11. Apoptosis Assay

We seeded A431 cells onto 12-well plates (1.2 × 10^5^ cells per well), transfected them with control siRNA, OVOL1 siRNA, or OVOL2 siRNA; and incubated them for 48 h at 37°C in 5% CO_2_. We determined the apoptotic status of the transfected cells using an Annexin V-FITC Apoptosis Detection Kit (15342-54; Nacalai Tesque Inc.) according to the manufacturer’s instructions. Briefly, we harvested the cells by trypsinization and suspended them in 1×Annexin V binding solution at 1.0 × 10^6^ cells/mL. We mixed 100 μL cell suspension with 5 μL Annexin V-FITC solution and 5 μL propidium iodide solution and then incubated them for 15 min at room temperature in the dark. Then, we mixed the cells with 400 μL 1×Annexin V binding solution and analyzed them using a FACSCanto II flow cytometer (BD Biosciences). We determined the proportion of FITC-positive apoptotic cells among live cells by using FlowJo software (663335; BD Biosciences). We conducted all assays in triplicate wells and repeated them at least three times in separate experiments.

### 2.12. Statistical Analysis

GraphPad Prism version 8 (GraphPad Software, San Diego, CA, USA) was used for all statistical analyses. We used Fisher’s exact test to analyze the relationship between OVOL1/2 expression and ZEB1 or vimentin expression in AK and cSCC tissues. We collected quantitative data from at least three independent experiments and measurements. The results are expressed as mean ± standard deviation. We determined differences between groups by the Mann–Whitney *U*-test. A *p*-value < 0.05 was assumed to indicate a statistically significant difference; * *p* < 0.05, ** *p* < 0.01, and *** *p* < 0.001.

## 3. Results

### 3.1. Immunohistochemistry

#### 3.1.1. OVOL1/2 Expression in AK and cSCC 

First, to clarify the clinical significance of OVOL1, OVOL2, ZEB1 and vimentin in AK and cSCC, we examined their expression in 30 AK, 30 cSCC, and 13 perilesional normal skin using immunohistochemical staining. Their representative hematoxylin and eosin staining was depicted in Appendix A. We calculated the percentages of OVOL1- or OVOL2-positive cells by counting tumor cells in three random fields. The expression of OVOL1 in normal skin and AK was comparable, whereas it was significantly decreased in cSCC (compared with normal; *p* = 0.0088 and AK; *p <* 0.0001) (Figure 1A, Figure 2A, and Appendix A). The expression of OVOL2 was upregulated in the nuclei of AK cells than those of normal keratinocytes (*p* = 0.0159) (Figure 1B, Figure 2B, Appendix A, and Appendix A). In AK, cells budding into the dermis, which were at the forefront of the EMT process, were strongly positive for OVOL2 (Figure 1B and Appendix A). Notably, the proportion of OVOL2-positive cells was significantly lower in cSCC than AK (*p* = 0.0006), but not than normal epidermis (*p* = 0.324) (Figure 2B). These results suggest that the loss of OVOL1 or OVOL2 protein may be associated with the transition from AK to invasive cSCC.

#### 3.1.2. ZEB1 and Vimentin Expression in AK and cSCC 

As ZEB1 and vimentin are important EMT-promoting molecules in SCCs [11,15], we examined ZEB1 expression in AK and cSCC. We considered samples to be positive for ZEB1 when nuclear expression was observed in more than 5% of tumor cells [26]. Most normal keratinocytes and AK cells exhibited faint or negative staining for ZEB1 in the examined specimens (Figure 1C and Figure 2C). In contrast, ZEB1 was overexpressed in many cSCC samples (compared with normal; *p* = 0.0143 and AK; *p* = 0.0061) (Figure 1C, Figure 2C, and Appendix A). 

We next examined vimentin expression. In normal epidermis, vimentin expression was not detected in keratinocytes and was only positive in Langerhans cells and melanocytes. We considered samples to be positive for vimentin when expression was observed in more than 10% of tumor cells [26]. Most AK cells exhibited faint or negative staining for vimentin in the examined specimens (Figure 1D and Figure 2D). In contrast to normal epidermis or AK, vimentin was overexpressed in many cSCC samples (compared with normal; *p* = 0.0081 and AK; *p* = 0.0048) (Figure 1D, Figure 2D, and Appendix A). 

#### 3.1.3. Associations between OVOL1/2 and ZEB1 or Vimentin in AK and cSCC

We subsequently analyzed the association between OVOL1/2 and ZEB1 or vimentin in AK plus cSCC clinical samples. We considered samples to have high OVOL1 or OVOL2 expression when the percentage of cells positive for either exceeded the median (i.e., 20% for OVOL1 and 40% for OVOL2) (Figure 2A,B). There was no significant relationship between OVOL1 and ZEB1 expression (Table 1). Moreover, there were no significant associations between vimentin and OVOL1 or OVOL2 expression (Table 2). On the other hand, tumors with low OVOL2 expression exhibited significantly higher ZEB1 expression than tumors with high OVOL2 expression (*p* = 0.0105; Table 1). We then evaluated the associations of OVOL1/2 with ZEB1 and vimentin expressions separately in AK and cSCC (Appendix A). OVOL2 with ZEB1 tended to show an association only in cSCC, but it did not reach the statistical significance (*p* = 0.0604) (Appendix A). Again, there were no significant associations among OVOL1 and ZEB1, OVOL1 and vimentin, OVOL2 and ZEB1, and OVOL2 and vimentin in AK (Appendix A). No association of OVOL1 and ZEB1, OVOL1 and vimentin, or OVOL2 and vimentin was evident in cSCC (Appendix A).

### 3.2. In Vitro Assays Using an A431 Cell Line

#### 3.2.1. Screening of EMT-Related Factors Influenced by OVOL1/2 Knockdown

To elucidate the molecular mechanisms between OVOL1/2 and EMT-related factors, we screened the factors potentially influenced by OVOL1/2. Accordingly, we used siRNA approach to analyze the mRNA levels of EMT-related factors—*E-cadherin*, *vimentin*, *ZEB1, ZEB2*, *Snail*, and *Twist1—*upon knockdown of either *OVOL1* or *OVOL2* in the A431 human SCC cell line. We confirmed *OVOL1* and *OVOL2* knockdown by qRT-PCR (Figure 3A). Among the EMT-related factors, *OVOL1* knockdown significantly increased the mRNA levels of *vimentin* (*p* = 0.0006) and *ZEB1* (*p* = 0.0262) compared to the control siRNA; there were no significant changes in the mRNA levels of *E-cadherin* (*p* = 0.163), *ZEB2* (*p* = 0.381), *Snail* (*p* = 0.902), or *Twist1* (*p* = 0.0717) (Figure 3B). Similarly, compared to the control siRNA, *OVOL2* knockdown significantly increased the mRNA levels of *vimentin* (*p* = 0.0012) and *ZEB1* (*p* = 0.0006) but significantly decreased that of *E-cadherin* (*p* = 0.0006); there were no significant changes in the mRNA levels of *ZEB2* (*p* = 0.709), *Snail* (*p* = 0.901), or *Twist1* (*p* = 0.0956) (Figure 3B). 

In addition, compared to A431 cells treated with control siRNA, cells treated with *OVOL1* or *OVOL2* siRNA showed a tendency to convert to a spindle-like mesenchymal morphology (Appendix A). These results indicated that *OVOL1* and *OVOL2* negatively regulate the EMT genes expression in A431 cells.

#### 3.2.2. Impacts of OVOL1/2 Knockdown on Vimentin, ZEB1, and E-cadherin Protein Levels

To further examine the roles of OVOL1 and OVOL2, we investigated the vimentin, ZEB1, and E-cadherin protein levels, which were changed in qRT-PCR, using western blotting upon OVOL1/2 knockdown. We confirmed the efficiency of OVOL1/2 knockdown at the protein level (Figure 4A, Appendix A). Compared to the control siRNA, OVOL1 knockdown significantly increased the protein levels of vimentin (*p* = 0.0286) and ZEB1 (*p* = 0.0286) (Figure 4B and Appendix A). Similarly, OVOL2 knockdown significantly increased the protein level of ZEB1 (*p* = 0.0286) (Figure 4B and Appendix A); it also increased the protein level of vimentin, albeit not significantly (*p* = 0.0571) (Figure 4B and Appendix A). OVOL1/2 knockdown did not alter the protein level of E-cadherin (*p* = 0.700/*p* = 0.999) (Figure 4B and Appendix A).

#### 3.2.3. Increased Invasion upon OVOL2 Knockdown 

Given that OVOL1/2 inhibited the expression of EMT-related factors such as vimentin and ZEB1, we hypothesized that knockdown of OVOL1 or OVOL2 may enhance tumor cell invasion. We simultaneously performed the migration (non-coated control insert) and invasion (Matrigel-coated Transwell insert) assay using A431 cells transfected with control siRNA, OVOL1 siRNA or OVOL2 siRNA (Figure 5A). Invasion ability was calculated by dividing the number of invaded cells through Matrigel-coated Transwell inserts by the number of migrated cells through non-coated control inserts. Notably, the invasion ability of OVOL2-knockdowned A431 cells was significantly increased compared with control siRNA-transfected counterparts (*p* = 0.0260) (Figure 5B). A431 cells transfected with OVOL1 siRNA tended to exhibit higher invasion ability than those transfected with control siRNA, however, it was not statistically significant (*p* = 0.394) (Figure 5B).

To investigate the other functional roles of OVOL1/2, we subsequently performed a wound healing assay (Figure 6A), proliferation assay (Figure 6B), colony formation assay (Figure 6C), and apoptosis assay (Figure 6D). There were no differences in cell migration status, cell proliferation, colony-formation ability, or apoptotic status between the controls and OVOL1-knockdowned or OVOL2-knockdowned A431 cells (Figure 6A–D). These results stress the particular importance of OVOL2 in attenuating the invasive ability of SCC cells. 

## 4. Discussion

EMT is considered as being able to determine the architectural arrangement of tissue on the basis of the formation of intercellular tight junctions and adherens junctions. EMT is also related to stromal invasion by tumor cells [7]. Furthermore, the expression of EMT markers is associated with carcinoma progression and metastasis. Regarding skin carcinoma, partial EMT, characterized by Twist1 expression without E-cadherin depletion, has been reported to be associated with the acquisition of invasive traits in SCC, although this process is downregulated in lymph node metastases [26]. Meanwhile, OVOL1 and OVOL2 are considered critical inducers of EMT/MET in human cancers [16]. Accordingly, inhibition of OVOL1 or OVOL2 induces complete EMT, whereas their overexpression leads to complete MET [27,28]. Thus, elevated OVOL2 expression might suppress hepatocellular carcinoma cell invasion and metastasis by restricting EMT [28]. Moreover, it has been reported that the OVOL2/ZEB1 circuit is related to epithelial regeneration and repair in adult mouse skin [29]. We previously proposed that the OVOL1/2 axis is an important modulator in cSCC [25]. While OVOL1 and OVOL2 have been described as gatekeepers that prevent mesenchymal transdifferentiation and maintain epithelial identity, their regulation is poorly understood. 

In the present study, knockdown of *OVOL1* or *OVOL2* increased the mRNA levels of *vimentin* and *ZEB1*. Moreover, OVOL1 knockdown significantly increased the protein levels of vimentin and ZEB1, while OVOL2 knockdown significantly increased that of ZEB1. Notably, both OVOL1 and OVOL2 knockdown did not affect the expression of E-cadherin; the downregulation of which is another epithelial marker for EMT [26,30]. Regarding this inconsistency, the EMT-associated traits of tumor cells might vary depending on the type of tissue and malignancy. Of note, partial EMT, which is characterized by the increased expression of mesenchymal markers without decreased E-cadherin expression, has been proposed as a novel concept of EMT and may explain our findings [26,30]. OVOL1/2 may be related to the “partial EMT”. In addition, *in vitro* functional assays demonstrated that loss of OVOL2, but not OVOL1, significantly augmented the invasive ability of SCC cells.

The importance of OVOL2/ZEB1 axis was supported by the immunohistochemical analysis using AK and cSCC samples. The expression of OVOL1 and OVOL2 were upregulated in AK and significantly downregulated in cSCC. In contrast, ZEB1 and vimentin were upregulated in cSCC, whereas most AK cells were negative or faintly stained for them, suggesting that downregulation of OVOL1/2 and upregulation of ZEB1 and vimentin may be associated with the progression of AK to cSCC. Further statistical analysis indicated the significant negative association of OVOL2, but not OVOL1, with ZEB1 expression. In addition, there was no significant relationship among OVOL1, OVOL2 and vimentin. These results collectively suggest that OVOL2 suppresses ZEB1 expression in human AK and cSCC and that the OVOL2/ZEB1 axis may play a crucial role in regulating the promotion of AK to cSCC. 

## 5. Conclusions

In conclusion, we found that OVOL2 might be an important modulator of EMT and the invasiveness of human SCC cells. Furthermore, we propose that the loss of OVOL2 and reciprocal upregulation of ZEB1 may be crucial for the promotion of AK to cSCC.

## Figures and Tables

**Figure 1 jcm-09-00618-f001:**
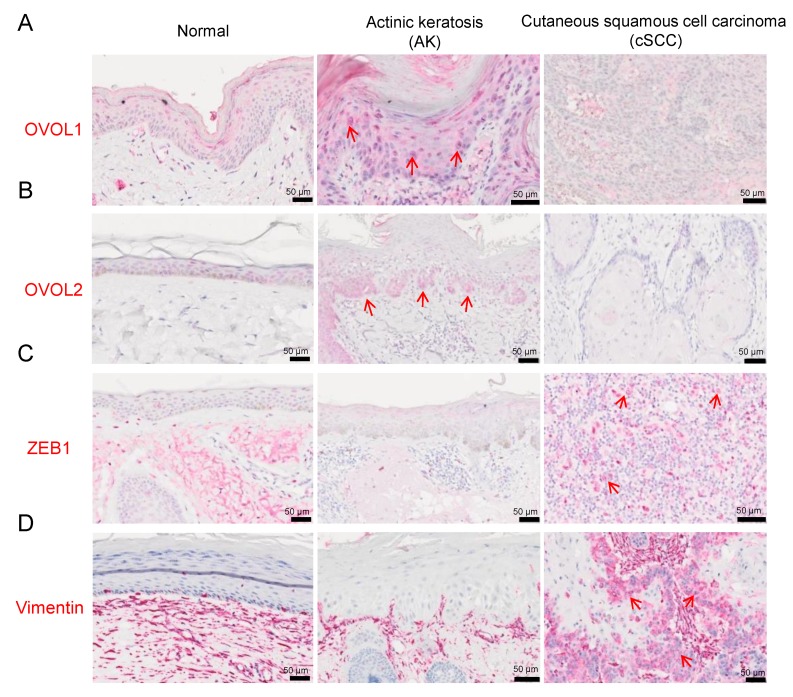
Representative images of ovo-like transcriptional repressor 1 (OVOL1) and ovo-like zinc finger 2 (OVOL2), zinc finger E-box binding homeobox 1 (ZEB1), and vimentin staining in normal skin (left), actinic keratosis (AK) (middle) and cutaneous squamous cell carcinoma (cSCC) (right). (**A**) Representative images of OVOL1 staining, (**B**) OVOL2 staining, (**C**) ZEB1 staining, and (**D**) vimentin staining. Red arrows show the positive cells.

**Figure 2 jcm-09-00618-f002:**
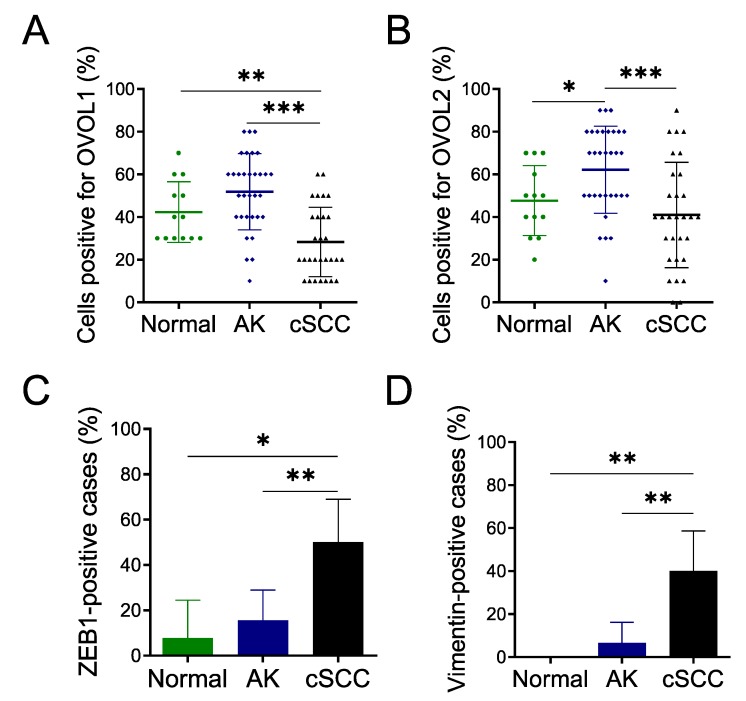
OVOL1, OVOL2, ZEB1, and vimentin exhibit opposite protein expression patterns between AK and cSCC. (**A–D**) OVOL1, OVOL2, ZEB1, and vimentin expression were examined in 30 AK, 30 cSCC, and 13 perilesional normal skins. **(A)** The percentage of cells positive for OVOL1 and **(B)** the percentage of cells positive for OVOL2. Dots in the bars in (**A**) and (**B**) represent each clinical sample. **(C)** The percentage of ZEB1-positive patient samples. Samples were considered positive for ZEB1 when nuclear expression was observed in more than 5% of tumor cells. **(D)** The percentage of vimentin-positive patient samples. Samples were considered positive for vimentin when expression was observed in more than 10% of tumor cells. Mann–Whitney *U*-test; error bars represent mean ± standard deviation. *p*-values < 0.05 were assumed to indicate a statistically significant difference; * *p* < 0.05, ** *p* < 0.01, and *** *p* < 0.001.

**Figure 3 jcm-09-00618-f003:**
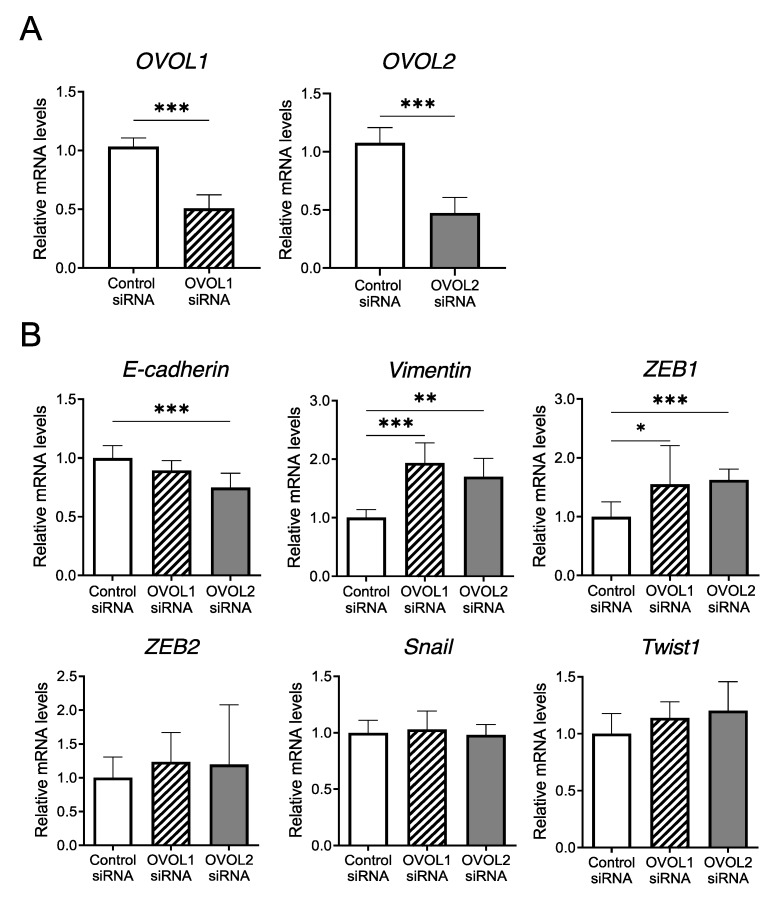
*Vimentin* and *ZEB1* mRNA levels are regulated by *OVOL1/2* knockdown in a human SCC cell line. (**A**) Relative mRNA levels of *OVOL1* and *OVOL2* in A431 cells treated with control siRNA, OVOL1 siRNA, or OVOL2 siRNA for 48 h. (**B**) Relative mRNA levels of *E-cadherin*, *vimentin*, *ZEB1*, *ZEB2*, *Snail*, and *Twist1* in A431 cells treated with control siRNA, OVOL1 siRNA, or OVOL2 siRNA for 48 h. Mann–Whitney *U*-test; error bars represent mean ± standard deviation. For interpretation purposes, the control siRNA value was set to 1. *p*-values < 0.05 were assumed to indicate a statistically significant difference; * *p* < 0.05, ** *p* < 0.01, and *** *p* < 0.001. Assays were conducted in triplicate and repeated at least three times in separate experiments.

**Figure 4 jcm-09-00618-f004:**
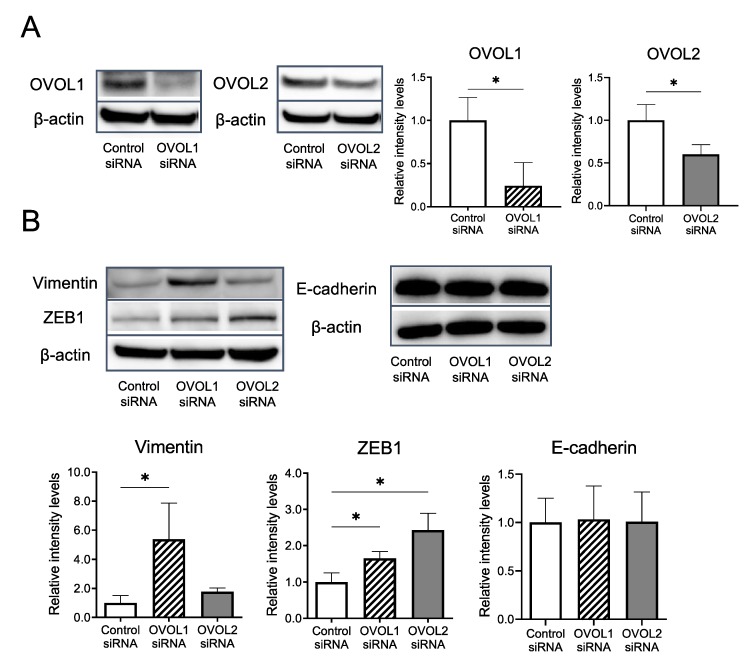
Vimentin and ZEB1 protein levels are regulated by OVOL1 and OVOL2 in a human SCC cell line. (**A**) Relative protein expression levels of OVOL1 and OVOL2 in A431 cells treated with control siRNA, OVOL1 siRNA, or OVOL2 siRNA for 48 h. (Left) Representative blot images and (right) relative expression levels calculated from three independent experiments. (**B**) Relative protein expression levels of vimentin, ZEB1, and E-cadherin in A431 cells treated with control siRNA, OVOL1 siRNA, or OVOL2 siRNA for 72 h. (Upper) representative blot images and (lower) relative protein expression levels calculated from three independent experiments. Mann–Whitney *U*-test; error bars represent mean ± standard deviation. Protein expressions are relative to those of β-actin as a reference. The control siRNA value was set to 1. *p*-values < 0.05 were assumed to indicate a statistically significant difference; * *p* < 0.05.

**Figure 5 jcm-09-00618-f005:**
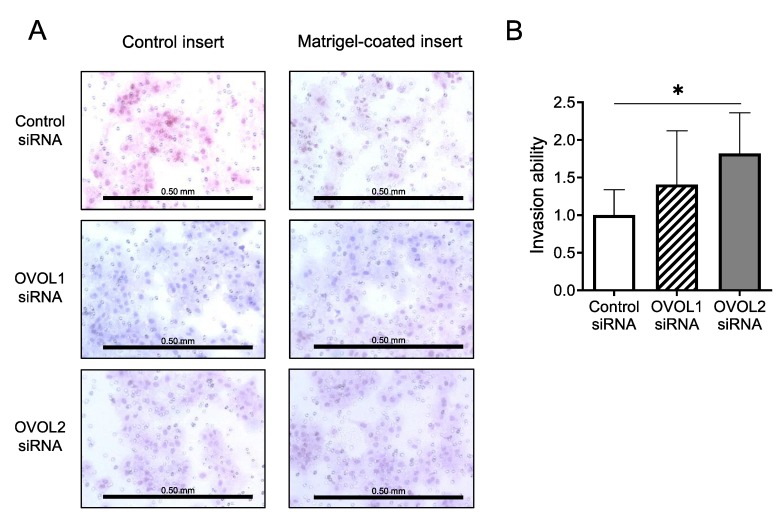
OVOL2 regulates invasion ability in a human SCC cell line. (**A**) Representative images, and (**B**) relative invasion ability of cells. A431 cells were transfected with control siRNA, OVOL1 siRNA, or OVOL2 siRNA and cultured in Transwell plates. Cells that translocated through control inserts or invaded Matrigel-coated inserts are stained on the membrane. Scale bar = 0.50 mm. Mann–Whitney *U*-test; error bars represent mean ± standard deviation. For interpretation purposes, the control siRNA value was set to 1. *p*-values < 0.05 were assumed to indicate a statistically significant difference; * *p* < 0.05. Assays were conducted in triplicate and repeated at least three times in separate experiments.

**Figure 6 jcm-09-00618-f006:**
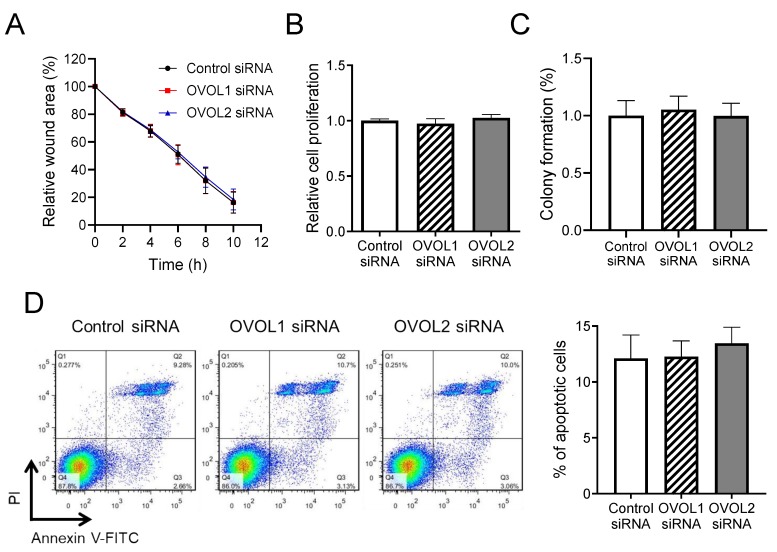
OVOL1/2 do not regulate wound healing, proliferation, colony formation, or apoptosis in a human SCC cell line. (**A**) A431 cells were transfected with control siRNA, OVOL1 siRNA, or OVOL2 siRNA and submitted to a wound healing assay. Forty-eight hours post-transfection, cell monolayers were scratched and tracked for cell migration until 10 h post-scratching. A representative result is shown. (**B**) Cell proliferation was examined in siRNA-transfected cells using the WST-1 assay 48 h post-transfection. (**C**) The colony-forming ability of siRNA-transfected cells was assessed using the semi-solid agar and MTT assay. (**D**) The apoptotic status of siRNA-transfected cells was evaluated by propidium iodide–Annexin V staining and measured by flow cytometry. (Left) Representative flow cytometric images and (right) the percentages of apoptotic cells in all populations. Mann–Whitney *U*-test; error bars represent mean ± standard deviation. *p*-values < 0.05 were assumed to indicate a statistically significant difference. All of the experiments were independently repeated at least three times.

**Table 1 jcm-09-00618-t001:** Associations between OVOL1/2 and ZEB1 expression in AK and cSCC clinical samples.

	Total	OVOL1		*p*-Value
		**High**	**Low**	
ZEB1				0.245
Positive	19	10	9	
Negative	41	29	12	
	**Total**	**OVOL2**		***p*-value**
		**High**	**Low**	
ZEB1				0.0105*
Positive	19	7	12	
Negative	41	30	11	

Fisher’s exact test. *p*-values < 0.05 were assumed to indicate a statistically significant difference; * *p* < 0.05; ZEB1, zinc finger E-box binding 1; OVOL1, ovo-like transcriptional repressor 1; OVOL2, ovo-like zinc finger 2.

**Table 2 jcm-09-00618-t002:** Associations between OVOL1/2 and vimentin expression in AK and cSCC clinical samples.

	Total	OVOL1		*p*-Value
		**High**	**Low**	
Vimentin				0.352
Positive	15	8	7	
Negative	45	31	14	
	**Total**	**OVOL2**		***p*-value**
		**High**	**Low**	
Vimentin				0.543
Positive	15	8	7	
Negative	45	29	16	

Fisher’s exact test. *p*-values < 0.05 were assumed to indicate a statistically significant difference.

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
