# Peer review of "OVOL2-Mediated ZEB1 Downregulation May Prevent Promotion of Actinic Keratosis to Cutaneous Squamous Cell Carcinoma"

_jcm, 2020, doi:10.3390/jcm9030618_

Round 1

Reviewer 1 Report

The manuscript by Murata et al. describes the influence of ovo like transcriptional repressor 1 (OVOL1) and ovo like zinc finger 2 (OVOL2) on factors relevant to epithelial-to-mesenchymal transition (EMT) relevant for the conversion of precancerous actinic keratosis lesions into cutaneous squamous cell carcinoma (cSCC). The study utilizes several techniques spanning from immunohistochemistry of patient samples to measurements of transcript and protein levels following siRNA-mediated knockdown to study expression levels of OVOL1/2 and their influence on EMT-related factors. A key finding is the role of OVOL2 to inhibit expression of zinc finger E-box binding homeobox 1 (Zeb1) and thereby inhibit development of sSCC. This relationship has been described previously and is also properly cited (e.g. refs 14-15).

The study is presented in a clear language with sufficient experimental details. The order of described experiments is logical and technical replicates as well as statistical analyses are clearly presented. There is generally a good correlation between observations made on the transcript level (e.g. by RT-PCR) and on the protein level (e.g. by Western blot). Overall the manuscript is sound and data presented in a thorough fashion.

I only have a few minor comments on the text:

Line 56: “changed” should be “change”. Line 77: “we found that OVOL2/ZEB1 axis…” should be “we found that the OVOL2/ZEB1 axis…”. The dilutions used for NBP1-88754 (1:100) and HPA027524 (1:200) in immunohistochemistry (1:100) are higher than the range recommended by the suppliers (1:500-1:1000), but that is probably something the experimentalists evaluated empirically. However, the 1:1 dilution used for antibody 722101 from Nichirei Biosciences appears high and I cannot find any details on this antibody online. Perhaps the authors can comment on this. Line 142: “and were performed transfected with siRNAs” should be “and transfected with siRNAs”. In the Western blot experiments (supplementary fig 3) the blots display significantly more smear/non-specific bands than in pictures shown by the suppliers of e.g. antibodies ab74520 and ab83265. Since the analysis is relative between control siRNA and OVLOV1/2 siRNA this is probably not a major issue. However, for OVLOV2 the expected band size according to Abcam should be ca 37 kDa whereas figure S3 indicates a size below 30 kDa. Can the authors please check this detail? Line 380: I assume “Ovol2/Zeb1” should be “OVOL2/ZEB1” to be consistent with the remainder of the text.

Author Response

Reply to the Reviewer 1

The manuscript by Murata et al. describes the influence of ovo like transcriptional repressor 1 (OVOL1) and ovo like zinc finger 2 (OVOL2) on factors relevant to epithelial-to-mesenchymal transition (EMT) relevant for the conversion of precancerous actinic keratosis lesions into cutaneous squamous cell carcinoma (cSCC). The study utilizes several techniques spanning from immunohistochemistry of patient samples to measurements of transcript and protein levels following siRNA-mediated knockdown to study expression levels of OVOL1/2 and their influence on EMT-related factors. A key finding is the role of OVOL2 to inhibit expression of zinc finger E-box binding homeobox 1 (Zeb1) and thereby inhibit development of sSCC. This relationship has been described previously and is also properly cited (e.g. refs 14-15). The study is presented in a clear language with sufficient experimental details. The order of described experiments is logical and technical replicates as well as statistical analyses are clearly presented. There is generally a good correlation between observations made on the transcript level (e.g. by RT-PCR) and on the protein level (e.g. by Western blot). Overall the manuscript is sound and data presented in a thorough fashion.

→ Thank you very much for your very encouraging comment.

I only have a few minor comments on the text:

Line 56: “changed” should be “change”.

→ We apologize our careless mistake. We amended it according to your comment (Line 58).

Line 77: “we found that OVOL2/ZEB1 axis…” should be “we found that the OVOL2/ZEB1 axis…”.

→ We apologize our careless mistake. We amended it according to your comment (Line 79).

The dilutions used for NBP1-88754 (1:100) and HPA027524 (1:200) in immunohistochemistry (1:100) are higher than the range recommended by the suppliers (1:500-1:1000), but that is probably something the experimentalists evaluated empirically. However, the 1:1 dilution used for antibody 722101 from Nichirei Biosciences appears high and I cannot find any details on this antibody online. Perhaps the authors can comment on this.

→ Thank you very much for your comment. The antibody 722101 from Nichirei Biosciences is a prediluted antibody by supplier. Thus we use it without further dilution. We amended this sentence as follows; (Prediluted by supplier, 722101; Nichirei Biosciences) (Line 99)

Line 142: “and were performed transfected with siRNAs” should be “and transfected with siRNAs”.

→ We apologize our careless mistake. We amended it according to your comment(Line 146).

In the Western blot experiments (supplementary fig 3) the blots display significantly more smear/non-specific bands than in pictures shown by the suppliers of e.g. antibodies ab74520 and ab83265. Since the analysis is relative between control siRNA and OVLOV1/2 siRNA this is probably not a major issue. However, for OVLOV2 the expected band size according to Abcam should be ca 37 kDa whereas figure S3 indicates a size below 30 kDa. Can the authors please check this detail?

→ Thank you very much for your comment.. According to your comment, we checked the supplier’s data sheet. The data sheet of ab83265 showed that the observed band size should be 30kDa. We attached a part of the datasheet and URL.

https://www.abcam.com/ovol2-antibody-ab83265.

Line 380: I assume “Ovol2/Zeb1” should be “OVOL2/ZEB1” to be consistent with the remainder of the text.

→ Thank you so much for your comment. We amended it according to your comment (Line 397).

Thank you very much again for your helpful comments. We hope the revised article is now suitable for publication in JCM.

Reviewer 2 Report

Comments to authors: I read the manuscript from Murata and Colleagues carefully. The overall goal of the study is interesting and further understanding how EMT is involved in cancer initiation and progression is important.

I take issue with the 1st sentence of the abstract and the framing of the manuscript. Actinic keratosis have been described as precancerous; but the rate of transformation to SCC is really low--0.1% to 0.6% per lesion per year (Criscone 2009, Werner 2015) with a high potential for spontaneous regression (Marks 1986). The sentence as written in the abstract should be modified to reflect that only an extremely low percentage of AKs progress to SCC and it is not as straightforward as it currently appears.   This language should also be altered throughout the entire paper as needed.  In addition, please supply/reference the appropriate studies demonstrating what percentage of SCC remain in situ vs those that metastasize.  Framing and context is key.

Methods: Tissue samples—were the 30 cSCC samples and 30 AK samples from 30 different individuals?  1 cSCC and 1 AK from each person or were the samples completely independent (no samples from the same person in both cSCC and AK groups). Please clarify.  Similarly, is the histological grade (low differentiation vs highly differentiated) of the cSCC samples known?  If yes, is there variation in the OVOL2 or OVOL1 protein expression or location?  Similarly—are there changes in ZEB-1 or Vimentin that is associated with tumor grade?

Figure 1: What is the expected staining intensity/pattern/number of cells in normal unaffected healthy skin for OVOL1, OVOL2, ZEB1, Vimentin?  Is there a difference between AKs and normal skin? Please supply normal skin data (n = 30?) and images. If the argument throughout this entire manuscript is that AKs are “precancerous” than expectations are that these markers are altered in the context of AKs compared to normal skin. These findings should be clearly described. In addition, please supply representative H&E images for normal skin, AKs and SCCs.  On the IHC panels, some please include some orientation notes—this is particularly necessary for the SCCs where it is impossible to discern epidermis/dermis boundary and orientation.

Table 1: The data as presented in Table 1 are confusing.  Are all samples, AKs and cSCCs lumped together? Why?  What is the breakdown of low/high expression for AKs and cSCCs.

Can the authors provide clarity as to the choice of cell line, A431, for the in vitro studies? A431 is a mucosal epidermal carcinoma cell line. Why not use HaCaT keratinocytes—at least this is derived from adult skin (mirroring the origin of AKs and cSCCs). Importantly if the argument is that OVAL1 and OVAL2 are already low in SCCs then wouldn’t something with normal levels to start be required?   Do these results repeat in other cells? It is important to demonstrate the rigor/reproducibility of all these studies in another cell line or normal primary keratinocyte cells (as a control).

Author Response

Reply to the Reviewer 2

Comments to authors: I read the manuscript from Murata and Colleagues carefully. The overall goal of the study is interesting and further understanding how EMT is involved in cancer initiation and progression is important.

→ Thank you so much for your encouraging comment.

I take issue with the 1st sentence of the abstract and the framing of the manuscript. Actinic keratosis has been described as precancerous; but the rate of transformation to SCC is really low--0.1% to 0.6% per lesion per year (Criscone 2009, Werner 2015) with a high potential for spontaneous regression (Marks 1986). The sentence as written in the abstract should be modified to reflect that only an extremely low percentage of AKs progress to SCC and it is not as straightforward as it currently appears. This language should also be altered throughout the entire paper as needed. In addition, please supply/reference the appropriate studies demonstrating what percentage of SCC remain in situ vs those that metastasize. Framing and context is key.

→ Thank you very much for your very critical comment. We totally agree with your comment.

According to your comment, we amended the Abstract and Introduction as follows citing your recommended articles (Ref 3 and 4);

Line 16

“Progression of actinic keratosis (AK) to cutaneous squamous cell carcinoma (cSCC) is rare.”

Line 39 to 44

“Most cases of AK remain intraepidermal lesions and the progression of AK to cSCC is rare [3, 4]. AK is a precursor of cSCC characterized by atypical keratinocytes within the epidermis. Early changes in AK start at the basal layer in the interfollicular epidermis [1, 5]. Although the rate of transformation of AK to cSCC is low, 0.1% to 0.6% per lesion per year, consequences of advanced cSCC can be devastating [3, 4]. Therefore, early-stage intervention and/or prophylaxis is essential.”

Methods: Tissue samples—were the 30 cSCC samples and 30 AK samples from 30 different individuals? 1 cSCC and 1 AK from each person or were the samples completely independent (no samples from the same person in both cSCC and AK groups). Please clarify.

→ Thank you so much for your comment. Our tissue samples were provided from 60 patients. According to your comment, we added the following sentences in Materials and methods (2.1. Tissue samples).

Line 87 to 88

“All samples were taken from completely independent patients. Thirteen perilesional normal skin in AK patients served as control.”

Similarly, is the histological grade (low differentiation vs highly differentiated) of the cSCC samples known? If yes, is there variation in the OVOL2 or OVOL1 protein expression or location? Similarly—are there changes in ZEB-1 or Vimentin that is associated with tumor grade?

→ Thank you very much for your important comment. However, most cSCC samples contained intermingled areas of differentiated and undifferentiated part, therefore, it was very difficult to grade. So we did not include differentiation grade analysis in this study. In the future study, we will try it after determining a good differentiation marker for cSCC.

Figure 1: What is the expected staining intensity/pattern/number of cells in normal unaffected healthy skin for OVOL1, OVOL2, ZEB1, Vimentin? Is there a difference between AKs and normal skin? Please supply normal skin data (n = 30?) and images. If the argument throughout this entire manuscript is that AKs are “precancerous” than expectations are that these markers are altered in the context of AKs compared to normal skin. These findings should be clearly described.

→ Thank you so much for your crucial comment. According to your comment, we included the 13 normal skin data (H&E, immunohistological and statistical data of OVOL1, OVOL2, ZEB1 and vimentin). We amended the revised article as follows;

Line 239 to 267

“3.1.1. OVOL1/2 expression in AK and cSCC

First, to clarify the clinical significance of OVOL1, OVOL2, ZEB1 and vimentin in AK and cSCC, we examined their expression in 30 AK, 30 cSCC and 13 perilesional normal skin using immunohistochemical staining. Their representative hematoxylin and eosin staining was depicted in Supplementary Figure S1. We calculated the percentages of OVOL1- or OVOL2-positive cells by counting tumor cells in three random fields. The expression of OVOL1 in normal skin and AK was comparable, whereas it was significantly decreased in cSCC (compared with normal; P = 0.0088 and AK; P < 0.0001) (Figure 1A, Figure 2A, Supplementary Figure S2A). The expression of OVOL2 was upregulated in the nuclei of AK cells than those of normal keratinocytes (P = 0.0159) (Figure 1B, Figure 2B, Supplementary Figure S2B, Supplementary Figure S3). In AK, cells budding into the dermis, which were at the forefront of the EMT process, were strongly positive for OVOL2 (Figure 1B, Supplementary Figure S3). Notably, the proportion of OVOL2-positive cells was significantly lower in cSCC than AK (P = 0.0006), but not than normal epidermis (P = 0.324) (Figure 2B). These results suggest that the loss of OVOL1 or OVOL2 protein may be associated with the transition from AK to invasive cSCC.

3.1.2. ZEB1 and vimentin expression in AK and cSCC

As ZEB1 and vimentin are important EMT-promoting molecules in SCCs [11,15], we examined ZEB1 expression in AK and cSCC. We considered samples to be positive for ZEB1 when nuclear expression was observed in more than 5% of tumor cells [26]. Most normal keratinocytes and AK cells exhibited faint or negative staining for ZEB1 in the examined specimens (Figure 1C, Figure 2C). In contrast, ZEB1 was overexpressed in many cSCC samples (compared with normal; P = 0.0143 and AK; P = 0.0061) (Figure 1C, Figure 2C, Supplementary Figure S2C).

We next examined vimentin expression. In normal epidermis, vimentin expression was not detected in keratinocytes and was only positive in Langerhans cells and melanocytes. We considered samples to be positive for vimentin when expression was observed in more than 10% of tumor cells [26]. Most AK cells exhibited faint or negative staining for vimentin in the examined specimens (Figure 1D, Figure 2D). In contrast to normal epidermis or AK, vimentin was overexpressed in many cSCC samples (compared with normal; P = 0.0081 and AK; P = 0.0048) (Figure 1D, Figure 2D, Supplementary Figure S2D). “

In addition, please supply representative H&E images for normal skin, AKs and SCCs. On the IHC panels, some please include some orientation notes—this is particularly necessary for the SCCs where it is impossible to discern epidermis/dermis boundary and orientation.

→ Thank you so much for your comment. According to your comment, we added representative H&E images of normal skin, AK and cSCC (Supplementary Figure S1). Moreover, we supplied a low-magnification field of view to clarify the location of cSCC tumor cells. Rectangular area in Supplementary Figure S2 were shown in higher power view in Figure 1.

Table 1: The data as presented in Table 1 are confusing. Are all samples, AKs and cSCCs lumped together? Why? What is the breakdown of low/high expression for AKs and cSCCs.

→ Thank you very much for your valuable comment. As you pointed, Table 1 showed the AK+cSCC analysis. According to your comment, we also separately analyzed the associations of OVOL1/2 with ZEB1 or vimentin in AK or cSCC in Supplementary Table S1A, Table S1B, Table S2A, and Table S2B. As described in the text, Only OVOL2 with ZEB1 in cSCC tended to show an association, but it did not reach the statistical significance (P = 0.0604). We considered samples to have high OVOL1 or OVOL2 expression when the percentage of cells positive for either exceeded the median. We added sentences in the revised article as follows;

Line 289 to 296

“We then evaluated the associations of OVOL1/2 with ZEB1 and vimentin expressions separately in AK and cSCC (Supplementary Tables S1A, S1B, S2A and S2B). OVOL2 with ZEB1 tended to show an association only in cSCC, but it did not reach the statistical significance (P = 0.0604) (Supplementary Table S1B). Again, there was no significant associations among OVOL1 and ZEB1, OVOL1 and vimentin, OVOL2 and ZEB1, and OVOL2 and vimentin in AK (Supplementary Tables S1A and S2A). No association of OVOL1 and ZEB1, OVOL1 and vimentin, or OVOL2 and vimentin was also evident in cSCC (Supplementary Tables S1B and S2B).”

Line284 to 286

“We considered samples to have high OVOL1 or OVOL2 expression when the percentage of cells positive for either exceeded the median (i.e., 20% for OVOL1 and 40% for OVOL2) (Figure 2A, 2B).”

Can the authors provide clarity as to the choice of cell line, A431, for the in vitro studies? A431 is a mucosal epidermal carcinoma cell line. Why not use HaCaT keratinocytes—at least this is derived from adult skin (mirroring the origin of AKs and cSCCs). Importantly if the argument is that OVOL1 and OVOL2 are already low in SCCs then wouldn’t something with normal levels to start be required? Do these results repeat in other cells? It is important to demonstrate the rigor/reproducibility of all these studies in another cell line or normal primary keratinocyte cells (as a control).

→ Thank you very much for your very critical comment. HaCaT keratinocytes are obviously immortal but remains nontumorigenic (Boukamp1988, see below). In addition, in our previous studies, they behave like normal human keratinocytes in terms of differentiation marker such as filaggrin and involucrin expression (Kiyomatsu-Oda et al, Ulziu et al. see below). Therefore, they are used as substitute for normal human keratinocytes in most studies. We could not find out good SCC cell line other than A431. But we would like to confirm the present results in other SCC cell lines in future studies. In normal keratinocytes, we have previously shown that OVOL1 is an upstream regulator of OVOL2 (Ito et al, see below), so that the situation seems to be more complex. As our final goal is to find out the strategy to convert (or normalize) cSCC cells to AK cells or near-normal cells, we selected A431 SCC cells but not HaCaT cells in the present study. We could not find out other commercially available cSCC cells this time. We agree with your comment and we will continue the OVOL1/OVOL2 experiments using different experimental conditions. Thank you so much again.

Boukamp P, Petrussevska RT, Breitkreutz D, Hornung J, Markham A, Fusenig NE. Normal keratinization in a spontaneously immortalized aneuploid human keratinocyte cell line. J Cell Biol. 1988 Mar;106(3):761-71.

Kiyomatsu-Oda M, Uchi H, Morino-Koga S, Furue M. Protective role of 6-formylindolo[3,2-b] carbazole (FICZ), an endogenous ligand for arylhydrocarbon receptor, in chronic mite-induced dermatitis. J Dermatol Sci. 2018 Jun;90(3):284-294.

Ulzii D, Kido-Nakahara M, Nakahara T, Tsuji G, Furue K, Hashimoto-Hachiya A, Furue M. Scratching Counteracts IL-13 Signaling by Upregulating the Decoy Receptor IL-13Rα2 in Keratinocytes. Int J Mol Sci. 2019 Jul 6;20(13). pii: E3324.

Ito T, Tsuji G, Ohno F, Nakahara T, Uchi H, Furue M. Potential role of the OVOL1-OVOL2 axis and c-Myc in the progression of cutaneous squamous cell carcinoma. Mod Pathol. 2017 Jul;30(7):919-927.

Thank you very much again for your helpful comments. We hope the revised article is now suitable for publication in JCM.

Round 2

Reviewer 2 Report

The authors addressed most of my comments in a satisfactory manner.